# Developing objective tools to study rock hyrax (*Procavia capensis*) behaviour in the field

Héloïse Brotier[1], Pablo Alba-Gonzalez[2], Prameek Kannan[1], Eli Geffen[2] Amiyaal Ilany[2,3], Lee Koren🆔[1,4]*

1 Faculty of Life Sciences, Bar Ilan University, Ramat Gan, Israel, 2 School of Zoology, Faculty of Life Sciences, Tel Aviv University, Tel Aviv, Israel, 3 The Steinhardt Museum of Natural History, Tel Aviv University, Tel Aviv, Israel, 4 The Gonda Multidisciplinary Brain Research Center, Bar Ilan University, Ramat Gan, Israel

* Lee.Koren@biu.ac.il

## Abstract

Animal behaviour research seeks methodological rigor and consistency within and across studies, yet maintaining shared standards requires continual evaluation. The rock hyrax (*Procavia capensis*) is a social mammal whose behaviour has been widely studied, but no standardized methodological ethogram currently exists for this species. Here, we developed an ethogram for the rock hyrax using a procedure designed to produce objective and reproducible behavioural classifications. We observed and filmed four groups of rock hyrax using focal animal sampling and constructed the ethogram based on two organizational rules: the level of behavioural description and the behavioural units. We identified 42 distinct behaviours and grouped them into six behavioural categories based on context. This grouping was statistically validated using correspondence analysis, multidimensional unfolding, and a cluster variables algorithm. In addition, we performed a biological validation by comparing seasonal behavioural patterns with those reported in previous studies of this system. The resulting ethogram, together with its validated behavioural categories, provides a standardized framework that can improve objectivity and consistency in future studies of rock hyrax behaviour. The approach described here is broadly applicable to other species and may facilitate comparative research across taxa.

## Introduction

Language uses specific words to construct sentences and transfer information. While each word is defined, the meaning of sentences depends on context, which may be subjective and influenced by the experience and emotional state of both the writer or speaker and the reader or listener [1]. Similarly, the study of animal behaviour requires the creation of standardized terms and detailed descriptions of behaviours, a convention known as an ethogram [2]. The task of labelling and defining behaviours is particularly challenging in social species with complex behavioural repertoires. For

**Data availability statement:** All relevant data are within the manuscript and its Supporting Information files.

**Funding:** This research was funded by Yad Hanadiv (grant number 10992) as part of an interdisciplinary funded project on the influence of scientists' own thoughts and implicit biases on the process of scientific discovery. The funders had no role in study design, data collection and analysis, decision to publish, or preparation of the manuscript.

**Competing interests:** The authors have declared that no competing interests exist.

example, "open mouth" has been described either as a distinct behaviour [3] or as one element within the broader description of attack behaviour [4].

It can be useful to divide behaviours into static (e.g., standing, sitting) versus dynamic (e.g., walking, jumping) [5–7], or into states (i.e., behaviours whose duration can be measured) versus events (i.e., instantaneous behaviours whose duration is difficult to measure) [2]. Static behaviours are typically classified as states, whereas dynamic behaviours are classified as events, such that one classification scheme is often sufficient. Furthermore, the level of descriptive complexity should be consistent throughout an ethogram. For example, walking can be described as a dynamic posture [7], a slow type of locomotion [8], or as a series of limb positions—head forward, neck intermediate, legs intermediate, and toes alternating in forward and downward positions [9].

Ethograms may also group behaviours into categories based on function or context, a process that may introduce subjectivity [8,10–13]. Thus, objectively categorizing and interpreting behaviour remains a central challenge in behavioural research and can benefit from tools such as statistical similarity [3] and analyses of behavioural sequences [14]. For example, based on statistical similarity, grimace can be considered a submissive behaviour if grouped with retreat and ears back [3]. Similarly, if behaviours that occur sequentially belong to the same category, behaviours preceding a fight may be considered aggressive [15].

Rock hyraxes (*Procavia capensis*; hereafter hyrax) are social, diurnal mammals [16] whose behaviour has been widely studied since the 1960s, yet no standardized ethogram exists for this species. Each study defined its own set of focal behaviours, resulting in a total of 24 behaviours across studies (Table SI1), often using different definitions for the same behaviours. The purpose of this study was to create a standardized ethogram, comparable to those developed for well-established model species (e.g., zebrafish *Danio rerio*) [4], for use by rock hyrax researchers.

In this study, we describe a structured process for developing an objective ethogram, which may also be applicable to other species. We first defined behaviours as static, dynamic, or complex (i.e., constellations composed of multiple behaviours). For each behaviour, we described the position of the trunk, legs, and head. We then classified behaviours according to context and tested the objectivity of this categorization using three approaches: correspondence analysis, multidimensional unfolding, and the cluster variables algorithm. Finally, we biologically validated the categorization by comparing the temporal occurrence of behavioural categories across sampling months with seasonal patterns reported in the literature (Table SI1).

## Materials and methods

### Species, study site and field procedures

The rock hyrax is widespread across Africa and the Middle East in rocky terrain [16–17]. It lives in crevices that provide protection from predators and environmental conditions [17–18]. The typical hyrax group includes a resident male, adult females, juveniles and pups [18–19].

This study is part of the long-term hyrax study, ongoing since 1999 in the Ein Gedi Nature Reserve (31°28′N, 35°24′E), west of the Dead Sea in the Judean Desert, Israel. Hyraxes have been studied for six months each year (March to August). In the 2024 field season, four social groups were observed (Table SI2) five days a week, in the mornings, beginning before sunrise and continuing until all hyraxes were either in shelter or inactive for 30 minutes. All groups included pups that were born in March. All hyraxes in the study were captured using live box traps (Tomahawk Live Trap Co., Tomahawk, WI, USA), tagged with a unique collar and/or earrings to allow behavioural observations [20]. We observed hyrax using focal animal sampling [21], which records all behaviours of a focal individual over a pre-established period. In this study, we observed and filmed hyrax behaviour using a video camera (Nikon CoolPix P1000) for 10 minutes or until the focal individual moved out of sight (following [22]).

## Ethogram rules

To create a consistent ethogram, we described the behaviour on two levels:

1. We defined the behaviour as static (e.g., stand, sit, lie) or dynamic (e.g., run, walk, jump) [6–7]. Complex behaviours that combined several static or dynamic behaviours were defined as constellations (e.g., chase, play, nurse) [13].

2. For each static or dynamic behaviour, we specified the position of three body parts: the trunk, the legs and the head in relation to the ground (Fig 1) [5,6,9].

## Behavioural categories

Videos (193±258 seconds, 2.3±2.6 behaviours) were manually annotated using the BORIS software [23]. Using the data from BORIS, we constructed a binary matrix where the columns are the presence or absence of behavioural constellations during a session, and the rows are the observation sessions. All behaviours were given an equal weight in each session. We assigned each session to one of six behavioural categories (affiliative, agonistic, foraging, mating, social, vocalizations) by the context and our experience with the study species. To evaluate the robustness of these categories we used three methods. First, we used correspondence analysis (CA) and the presence/absence matrix of behavioural constellations to test for association between behavioural categories and constellations. Specifically, we aimed to identify

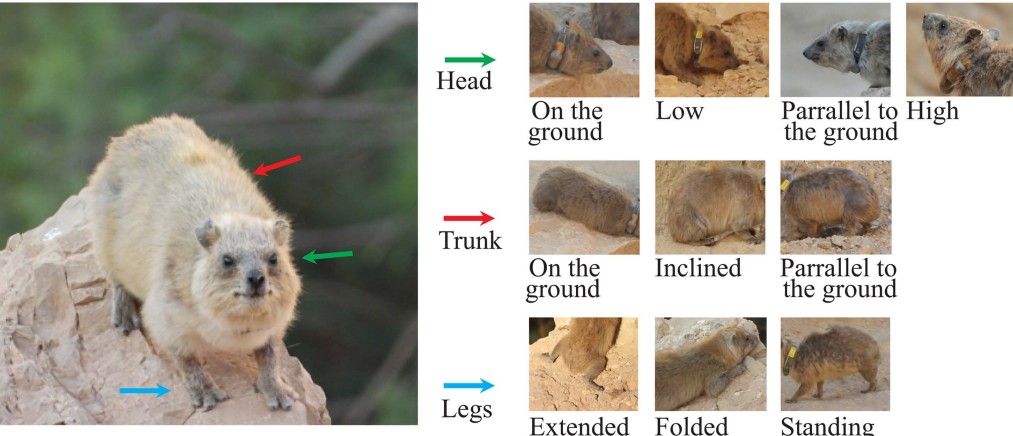

**Fig 1. Terms attributed to the three body parts positions.** Colours denote the different body parts and pictures illustrate the positions. The positions assigned to the head: on the ground, low, parallel to the ground, or high. The positions assigned to the trunk: on the ground, inclined, or parallel to the ground. The positions assigned to the legs: extended (to the side/behind), folded (below the trunk), or standing.

behavioural constellations that are associated with more than a single behavioural category. CA is conceptually like principal component analysis (PCA) for categorical (i.e., presence/absence) rather than continuous data. Then, we used two multivariate ordination procedures to classify constellations into clusters using the presence/absence matrix of behavioural constellations in the 156 video sessions. In the first approach we calculated the Jaccard similarity distance and the Bray-Curtis dissimilarity distance between the 23 behavioural constellations in SPSS (version 29, IBM Inc.). These two distance measures are the most frequently used for binary data. Using multidimensional unfolding (PREFSCAL, SPSS), we projected the Jaccard and Bray-Curtis distance matrices onto two-dimensional space. The stress measure (SM) and the dispersion accounted for (DAF) were used for evaluating the fit of projection of the two distance matrices. In the second approach, we used the cluster variables algorithm, which is a dimension-reduction method that constructs clusters by linear combinations of variables through a series of iterative steps and orthoblique rotation using SAS PROC VARCLUS [24], implemented in JMP Pro (version 18, SAS Institute Inc.). Unlike principal component analysis (PCA), this algorithm iteratively splits clusters of variables and reassigns variables to clusters until no more splits are possible. The cluster components produced in this analysis are not orthogonal but explain as much of the variation as possible among the variables within each cluster [24].

To examine the temporal congruency between previous studies of hyrax behaviour (Table SI1) and the behavioural categories in our ethogram, we counted the occurrences of the seven behavioural categories per month (March, April, May, June, July) for the 2024 field season. Then, we tested the association between each behavioural category and month using contingency table exact chi-square statistics and correspondence analysis (CA) in JMP (Version 18 Pro; SAS Institute Inc., USA).

### Ethical note

The hyrax research is conducted under annual permits from the Israeli Nature and Parks Authority (NPA) for capturing, handling and tagging at the Ein Gedi Nature Reserve (2024/43519). We have been using collars to identify individual hyrax since 2000. Before starting to use them on the study population we observed hyrax social behaviour with and without the collars on captive hyraxes and found that the collars do not change the behaviour. All procedures performed in the hyrax research in general are in accordance with the ethical standards of the ARRIVE guidelines. This specific study does not include experimental procedures. It involves behavioural observations from a distance (~ 50 meters) using binoculars, telescope, and a video camera.

## Results

### Ethogram

Using 1528 behaviour videos, we described 7 static and 11 dynamic behaviours (Table 1). Based on these, we defined 24 constellations, which we grouped into 6 categories (Table 1): affiliative behaviours (n=4), foraging behaviours (n=3), agonistic behaviours (n=4), social behaviours (n=3), mating behaviours (n=6), and vocalization (n=3). An additional behaviour (sand bathing) was categorized as miscellaneous.

### Association between behavioural constellations and behavioural categories

Correspondence analysis (CA) based on the frequency of occurrence of each constellation over all events and each of the seven behavioural categories showed that the first three coordinates (i.e., vectors) accounted together for 67.6% of the total variance (Fig 2a - 2c). CA showed a significant and clear association between constellations and the assigned behavioural categories (n=284, $\chi^2_{132}$=899.1; P<0.0001) except for two cases (social and agonistic and affiliative and miscellaneous categories). In all three CA plots, the same behavioural constellations are shared for the social and agonistic categories, and the same behavioural constellations are shared for the affiliative and miscellaneous categories (Fig 2a - 2c).

**Table 1. Rock hyrax ethogram: static positions, dynamic positions and behavioural constellations. For each behaviour there is a definition and a link to a picture or a short video with an example of it.**

| Type | Behaviour | Definition |
|---|---|---|
| Static behaviours | Lie 1 | The entire trunk and head rest on the ground, all four legs are extended backward and sideways. The body is completely in contact with the ground. [https://youtu.be/ylpakOpIc4Y] |
| | Lie 2 | The trunk is on the ground, the front legs are folded beneath the body and the back legs are extended backward and sideways. The head is either in a parallel to the ground position or resting on the ground. [https://youtu.be/R8RKsRAV6Rw] |
| | Lie 3 | The trunk touches the ground, and all legs are folded beneath the body. The head remains in a parallel to the ground position. [https://youtu.be/8yrW1JSfBYs] |
| | Sit | The front legs are stretched, and the back legs are folded beneath the body. The trunk is inclined, and the hindquarters rest on the ground. The head can be in a parallel to the ground, high, or low position. [https://youtu.be/m7wIyqx0mF0] |
| | Stand | Both front and back legs are in an erect position. The trunk remains parallel to the ground without touching it, and the head is held in a parallel to the ground position. [https://youtu.be/KqIIrFYkA4E] |
| | Dorsal hair erection | The hair around the dorsal gland stands upright. Often occurs during fights, chases, approaches, or when singing. [https://youtu.be/PqJep7TMBAQ] |
| | Penis erection | The penis is erect, typically during the mating season, when approaching a female. [https://youtu.be/F9A2IegMCMA] |
| Dynamic behaviours | Walk | An alternating pattern of legs movement called lateral couplets, where each leg moves distinctly. The head is in a parallel to the ground position, and the trunk remains parallel to the ground. The movement is steady and allows traveling from one point to another at a constant rate. [https://youtu.be/fsnPbvjnbOw] |
| | Run | A rapid alternating movement, where both back legs move together, followed by both front legs, in a pattern called bounding. The head remains in a parallel to the ground position, the trunk parallel to the ground. [https://youtu.be/mcjNDnLnSnk] |
| | Flee | Run from a threat or visible predator. Running continues until retreat into shelter under a rock or tree or at a distance from the source of disturbance. [https://youtu.be/Rhh6DjiI0vo] |
| | Jump | Both back legs are first folded, then extends to standing position to launch the body onto another location. The head is held in a parallel to the ground or high position, while the trunk remains parallel to the ground throughout the movement. [https://youtu.be/G_c0-6l9ZbY] |
| | Fall | Losing balance during a dynamic movement (i.e., run, walk, or jump). [https://youtu.be/UhDLHChshZI] |
| | Head shake | A brief, rapid side to side movement. This can occur in a variety of static positions such as stand, sit, lie 2, or lie 3, with the head in any position. [https://youtu.be/j11F6R4jmUU] |
| | Yawn | Jaws open slowly, stay open for a few seconds at maximum extension, and then close. [https://youtu.be/yh_mv0L_AMI] |
| | Chew | Repeated up-and-down jaws movements, mimicking chewing even when no food is present. [https://youtu.be/n9wJRv7gEFE] |
| | Scratch with teeth | Turning head at a 90-degree angle to reach the back, then repeated opening and closing movements of the mouth. The teeth contact the skin, in lie 2, lie 3, stand, or sit position. [https://youtu.be/dOUuhbIfNmA] |
| | Scratch with back paws | Moving the back paws up and down against the trunk or head. Typically occurs in a stand position. [https://youtu.be/czQVX3cziLk] |
| | Scratch with an object | Rubbing head or trunk against an object (e.g., a rock or a tree). Often repeated several times, with short breaks, and in sit or stand positions. [https://youtu.be/w5R2OYY-rXQ] |
| **Category** | **Constellation** | **Definition** |
| Affiliative behaviours | Babysit | Following and staying close to a pup. Mainly by females that are mothers. When pups are very young, multiple adults may participate in babysitting, including juveniles and resident males. [https://youtu.be/i6JVTGiWQ3w] |

*(Continued)*

| Type | Behaviour | Definition |
|------|-----------|------------|
| | Nurse/Suckle | Female stands while a pup places its head under her belly or chest. Once a pup attaches to a nipple, brief up and down head movement while suckling. Females nurse while pups suckle. A pup may attempt to suckle from a juvenile or a male. [https://youtu.be/V6hrt1HmGDM] |
| | Social touch | Two or more hyraxes rest in close contact, with at least one body part touching in a static position (e.g., sit, lie 1, lie 2, or lie 3). [https://youtu.be/4ZP2DuHf73k] |
| | Play | Series of short constellations such as chase, follow, mount, and bite. This constellation is primarily seen in pups and declines with age. [https://youtu.be/Y4Rct3PCvsI] |
| Foraging behaviours | Browse | Feeding on tree branches and leaves in a stand position, sometimes rising onto hind legs and using front legs to reach higher branches. The head is positioned low, parallel to the ground, or high, depending on the height of the food source. [https://youtu.be/OaIJSzu6xjc] |
| | Graze | Feeding on ground vegetation (e.g., herbs, small bushes or dry leaves). The body stays in a stand position, head in low position and swiping along the ground, and the hyrax may walk while grazing. Unlike browse, graze does not involve using the front paws to manipulate the food. [https://youtu.be/mNd8vMRdwyY] |
| | Coprophagy | Feeding on faeces on the ground. This constellation is identical to grazing. Once faeces are found, the hyrax remains in a standing position to consume it. [https://youtu.be/TxcoxgHKUUU] |
| Miscellaneous | Sand bath | Rolling from side to side on sandy ground while lying on the back. [https://youtu.be/Ws2XvdNBq8U] |
| Agonistic behaviours | Displacement | Run or walk towards an individual in a static position that moves away to a new static position. The displacing hyrax then takes the place of the displaced. [https://youtu.be/-p875lHo3uA] |
| | Chase | Run at maximum speed behind another running individual. When stops, remains standing with the head directed toward the fleeing individual, while it continues running. Usually ends when the fleeing hyrax reaches a safe place or stops at a distance. [https://youtu.be/TQHiPJ40dfw] |
| | Bite | Opening of the mouth and sinking of the teeth into another individual. Both hyraxes are often in a stand position. Occurs during fights. [https://youtu.be/Az_IVKN66_4] |
| | Fight | A series of aggressive actions (e.g., chase and bite). Physical contact occurs, with one hyrax emerging as the winner (e.g., the chaser, displacer, or aggressor) and the other as the loser (e.g., the one chased or displaced). [https://youtu.be/GtNfosh7rsQ] |
| Social behaviours | Sniff | Snouts contact with the ground, the body, or the rear of another hyrax. Body is in stand or sit positions with head in a low or parallel to the ground position, fixed or in swiping motion. [https://youtu.be/tUWzE3kameg] |
| | Follow | Movement in the same direction and at the same speed, by either walk or run. Hyraxes often stop at the same place, in any static position (stand, sit, lie 1, lie 2 or lie 3). [https://youtu.be/H9wMIiTvzLg] |
| | Join | Walk or run to join one or more hyraxes, which are in static positions. [https://youtu.be/23tfOSwk9R0] |
| Mating behaviours | Approach | Slow walk towards another hyrax of the opposite sex. Often accompanied by arousal signs such as dorsal hair or penis erection. May be followed by mount, copulation, or aggression. [https://youtu.be/2-1mqo9J0S0] |
| | Present | A female in a standing position with its head in a parallel to the ground position presents her hindquarters to a male. If the male moves away, the female may reposition herself to maintain presentation. This behaviour often leads to mounting and copulation. [https://youtu.be/YztbXDhUObc] |
| | Mount | Placement of male trunk over another's hindquarters. If mating-related, the male mounts a female, and his penis is usually erect. This constellation is often followed by copulation. If dominance-relate, two males mount each other, which may escalate to aggressive behaviour. [https://youtu.be/-HiJZhzZKPM] |
| | Copulate | Mount and penetration of a female by a male. The penis is erect, and thrusting occurs. After copulation, the female often bites or attempts to bite the male. [https://youtu.be/LG2Dim7tJy0] |
| | Mate rejection | Avoidance of an approach or presentation by moving away or engaging in aggression. May be done by a female or a male. [https://youtu.be/E-7aC4Qig24] |
| | Back-to-back | Backs facing in opposite directions while sitting or standing. [https://youtu.be/Tzhx7kk4Ygg] |

*(Continued)*

**Table 1.** (Continued)

| Type | Behaviour | Definition |
|---|---|---|
| Vocalizations | Sing | Modulated vocalization accompanied by repeated opening and closing movements of the jaws. The back hair is typically erect, the head is in a parallel to the ground position, and the body is in any static position. [https://youtu.be/dG1MrRnPkMM] |
| | Alarm call | Short emission of high-pitched, loud trill while in a stand, sit, or lie 3 positions. Often repeated and may follow the appearance of a predator, or a chase. [https://youtu.be/Z85JLkCj1MI] |
| | Pup screams | High-pitched screams when pup is directly threatened. May be brief or prolonged. This vocalization attracts the attention of nearby adults and may trigger singing. [https://youtu.be/8NiVBNALGmg] |

## Classification of behavioural constellations

We attempted to cluster the behavioural constellations based on resemblance of occurrence using two approaches: multidimensional unfolding and the cluster variables algorithm. The multidimensional unfolding plot for both distance measures showed a very low stress values (0.04) and nearly all the variance in the distance matrices was accounted for in the plots (DAF = 0.96; Fig 2d - 2e). Both plots showed a separate clustering for the mating and agonistic constellations. However, the constellations assigned to affiliative, foraging, miscellaneous, social, and vocalizations were not clustered by behavioural categories but formed two clusters of a mix of constellations that are associated with social, affiliative, and foraging (Fig 2d - 2e).

The Cluster Variables SAS PROC VARCLUS algorithm constructed a more structural set of clusters (Fig 2f), which accounted for 51% of the variance in the data (S1 Fig in S1 File). Cluster 1 is a set of constellations associated with mating behaviour. Cluster 2 clumped all the constellations associated with agonistic behaviour. Cluster 3 is a collection of behaviours related to maternal care. Foraging behaviours are united under Cluster 4. Cluster 5 and 7 combined behaviours that are related to stressful situations in adults and pups, respectively. Finally, Cluster 6 combined behaviours that are associated with male competition.

## Temporal analysis of behavioural constellations

During the 2024 field season, parturition spanned late March to April, followed by extensive maternal care (Fig 3a). Maternal care gradually decreased throughout the field season, which ended in late July. Matings started in late June, with mating-related behaviours increasing in frequency to the end of July (Fig 3a). The contingency table showed a significant association between behavioural category and month (n = 1785, $\chi^2_{24}$ = 220.9, P ≤ 0.0001; Fig 3b). The CA showed an association between affiliative behaviours and the month of April, sand bath and the month of May, social, agonistic and foraging behaviours and the month of June, and mating and vocalization behaviours and the month of July (Fig 3b).

## Discussion

An ethogram is a fundamental tool for objective animal behaviour studies [25–26]. Here, we developed an ethogram for the rock hyrax, a species whose behaviour we have studied since 1999, and defined and classified 42 behaviours according to strict organizational rules. These behaviours were grouped into six behavioural categories, which were validated using three statistical approaches. All three methods provided a good representation of the data, indicating that the resulting categorization captured the main behavioural structure. The high proportion of explained variance in the correspondence analysis (CA) suggests that the extracted dimensions accounted for most of the behavioural variation. The clear association between constellations and behavioural categories indicates that the constellations were appropriately classified and that the predefined categories are representative of hyrax behaviour. The multidimensional unfolding analysis revealed clear clustering of mating and agonistic constellations. In contrast, constellations that were initially assigned to social, affiliative, and foraging categories were distributed across clusters dominated by vocalization and

**Fig 2. Multivariate analysis for behaviour constellations (blue circles) and categories (red crosses).** Correspondence analysis plots are provided for coordinates 1 and 2 **(a)**, coordinates 1 and 3 **(b)** and coordinates 2 and 3 **(c)**. The amount of variance explained by each coordinate is denoted in parenthesis. Multidimensional unfolding map for behavioural constellations based on Jaccard similarity distance (d) and Bray-Curtis dissimilarity distance

**(e)**. The measures of stress and dispersion accounted for (DAF) are provided in parenthesis. Clustering of behavioural constellations by the Cluster Variables SAS PROC VARCLUS algorithm **(f)**. Coefficient level for each behavioural constellation and cluster are denoted by colour and details are provided in the legend. The amount of variance explained within cluster and in total are provided in parentheses.

a

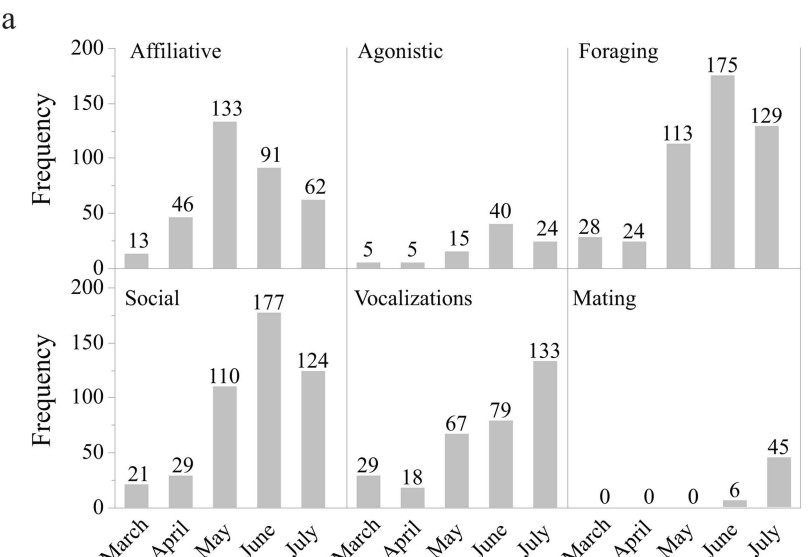

b

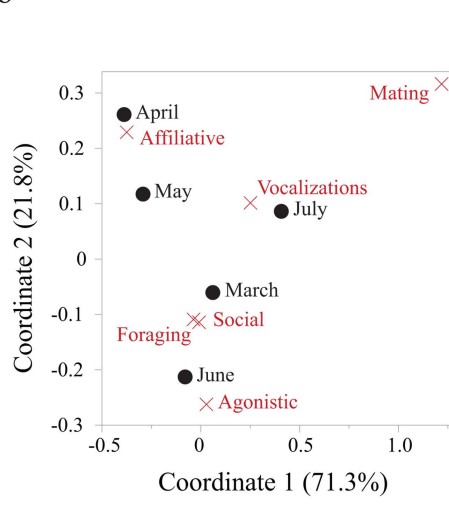

**Fig 3. Observations of behaviours by month. (a)** The frequency of occurrence (i.e., number of cases observed) for each of the seven behavioural categories by month during the 2024 field season. **(b)** Correspondence analysis showing the significant association ($\chi^{224} = 220.9$, $P \leq 0.0001$) between the behavioural categories (red cross) and months (blue circle). In parentheses the amount of variance explained by each coordinate.

agonistic behaviours. This pattern likely reflects the temporal overlap of social and foraging behaviours within the daily activity routine of hyraxes [27]. Similar results were obtained using the cluster variables algorithm, which also identified distinct clusters for mating and agonistic behaviours, while foraging, social, affiliative, and vocalization behaviours were distributed across other clusters. Across the three clustering approaches, mating emerged as the most robust behavioural category, followed by agonistic (supported by CA and the cluster variables algorithm), vocalization (supported by CA and the cluster variables algorithm), and foraging (highlighted by CA). In contrast, social and affiliative categories showed substantial overlap, and the clustering methods produced inconsistent results for these behaviours. By integrating the results of all three analyses, we were nevertheless able to assign each constellation to a single behavioural category with greater confidence. Consequently, the pre-established behavioural categories were revised: play and nursing behaviours were reclassified as affiliative behaviours, and back-to-back behaviour was reassigned from the agonistic to the mating category, as supported by the clustering analyses.

The temporal analysis provided a biological validation of the categorization, demonstrating that the seasonal patterns observed during the 2024 field season closely matched those reported in previous studies (Table SI1). For example, in 2024, parturition occurred in early April, and accordingly, babysitting, nursing, and affiliative behaviours declined from May onward, reaching low levels by the mating season in July. Nursing is expected to decline once pups are weaned at around 2–3 months of age, typically before the mating season [17]. We also observed that social and agonistic behaviours increased from March to June, consistent with patterns reported in the literature [17]. Mating behaviours were highly seasonal, showing a marked increase throughout the mating season and peaking in July, when mating activity was most prevalent [17,28]. Vocalizations also increased in July, coinciding with the mating season. Singing in hyraxes is thought

to signal male quality to rivals and potential mates [29–30]. Finally, foraging behaviours peaked between May and July in 2024, although comparable seasonal trends were not documented in earlier studies [17,27].

Overall, clustering results indicated that most behavioural categories were consistent with the literature and broadly supported across the three analytical approaches. However, some constellations were difficult to assign to a single category because they occurred in multiple contexts. For example, approach behaviour can occur in both mating and social contexts. In addition, some apparent inconsistencies may stem from differences in terminology across studies. Behaviours such as grooming [17,31] heaping [31], submissive behaviour [17], courtship [28], and vigilance [17,31,32] are not explicitly labelled in our ethogram, although the behaviours they describe are included. For instance, grooming represents a subjective interpretation (e.g., cleaning) and was therefore omitted as a label. Instead, the underlying behaviour: scratching with teeth, paws, or an object, is explicitly described. Courtship, previously defined as a male and a female approaching and following each other while sniffing and/or licking [28], comprises several constellations in our ethogram (approach, follow, sniff). This suggests that behavioural levels above constellations may exist, and that alternative hierarchical organizations of behavioural units are possible. Vigilance is another example of a subjective interpretation that we omitted. However, our ethogram includes the static behaviour stand with the head parallel to the ground, which captures the same mechanical behaviour without relying on contextual interpretation. Human observers are prone to assigning meaning to behaviours, which can result in over-interpretation and reduced objectivity.

In this study, we propose a procedure for defining and objectively testing the clustering of complex behaviours into categories, beginning with descriptions based on body-part positions and clearly defined behavioural units. Alternative approaches are possible, including the use of higher levels of descriptive complexity that incorporate additional body parts (e.g., mouth, eyes, nose, tail, and neck movements) [5,9]. Other methods, such as alternative statistical clustering techniques [3,33,34] free-choice profiling to assess behavioural intensity [35] or refined behavioural-unit frameworks with clearer boundaries [36], may further improve ethogram accuracy. Here, we present a comprehensive ethogram for wild rock hyraxes using a method designed to be simple and reproducible, thereby minimizing variability and enhancing objectivity. The ethogram was validated through temporal analyses, demonstrating its reliability. Together with the accompanying video material, this ethogram can be used to train new researchers and to assess inter-observer reliability prior to future studies. We suggest that this methodological framework may be applicable to other systems and broadly useful in animal behaviour research.

## Supporting information

**S1 File. Cluster Variables (SAS PROC VARCLUS) model results. Correlation color map for the 23 constellations.** Cluster assignment of all the constellations, R2 of each constellation with its own cluster, and with the next best cluster are listed on the left side of the map.
(ZIP)

## Acknowledgments

This research was part of an interdisciplinary project on the influence of scientists' own thoughts and implicit biases on the process of scientific discovery. Our team includes Galit Yovel (cognitive scientist from Tel Aviv University), Nurit Stadler (anthropologist from The Hebrew University), Naama Cohen-Hanegbi and Yuval Rotman (historians from Tel Aviv University) and the discussions were constructive for developing the framework for this manuscript. We are grateful to the Ein Gedi Nature Reserve and the Ein Gedi Field School staff for their hospitality.

## Author contributions

**Conceptualization:** Amiyaal Ilany, Lee Koren.

**Data curation:** Héloïse Brotier.

**Formal analysis:** Eli Geffen, Lee Koren.

**Funding acquisition:** Eli Geffen, Amiyaal Ilany, Lee Koren.

**Investigation:** Héloïse Brotier, Pablo Alba-Gonzalez, Prameek Kannan, Lee Koren.

**Methodology:** Héloïse Brotier, Eli Geffen, Amiyaal Ilany, Lee Koren.

**Project administration:** Amiyaal Ilany, Lee Koren.

**Resources:** Eli Geffen, Lee Koren.

**Software:** Eli Geffen.

**Supervision:** Amiyaal Ilany, Lee Koren.

**Validation:** Eli Geffen, Lee Koren.

**Writing – original draft:** Héloïse Brotier.

**Writing – review & editing:** Héloïse Brotier, Pablo Alba-Gonzalez, Prameek Kannan, Eli Geffen, Amiyaal Ilany, Lee Koren.

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
