## [Decision Letter · Decision Letter 0]

14 Jan 2026

Dear Dr. Koren,

Thank you for submitting your manuscript to PLOS ONE. After careful consideration, we feel that it has merit but does not fully meet PLOS ONE’s publication criteria as it currently stands. Therefore, we invite you to submit a revised version of the manuscript that addresses the points raised during the review process.

We look forward to receiving your revised manuscript.

Kind regards,

James Edward Brereton, MSc

Academic Editor

PLOS One

**Journal Requirements:**

“This research was funded by Yad Hanadiv (grant number 10992) as part of an interdisciplinary funded project on the influence of scientists’ own thoughts and implicit biases on the process of scientific discovery.”

4. We note that Figure 1 in your submission contain copyrighted images. All PLOS content is published under the Creative Commons Attribution License (CC BY 4.0), which means that the manuscript, images, and Supporting Information files will be freely available online, and any third party is permitted to access, download, copy, distribute, and use these materials in any way, even commercially, with proper attribution. For more information, see our copyright guidelines: http://journals.plos.org/plosone/s/licenses-and-copyright.

**Additional Editor Comments:**

Thank you for your patience during the review process. We have had challenged with recruiting reviewers for your study. At current, major revisions are required in order to consider the manuscript further. Below, the specific feedback comments are provided.

Reviewers' comments:

Reviewer's Responses to Questions

**Comments to the Author**

1. Is the manuscript technically sound, and do the data support the conclusions?

Reviewer #1: No

2. Has the statistical analysis been performed appropriately and rigorously?

Reviewer #1: No

3. Have the authors made all data underlying the findings in their manuscript fully available?

Reviewer #1: Yes

4. Is the manuscript presented in an intelligible fashion and written in standard English?

Reviewer #1: No

Reviewer #1: - Throughout the manuscript there are multiple uses of first person narrative, consider changing this to third person.

- The use of excessive wording at times obscures the main point and detracts from the scientific professionalism of the text. Simplifying the language would improve both clarity and impact and help maintain a professional and academic style.

- Several constellations (social vs affiliative vs foraging vs vocalisations) overlap across contexts and did not cluster cleanly, indicating limits to context-only categorical assignment for multifunctional behaviours.

- The presence/absence binary coding per session downplays intensity, duration, and frequency differences between behaviours within sessions, potentially meaning important variations get lost.

-Videos were relatively short on average and sessions ended when focal individuals moved out of view, which can bias detection of longer-duration states or rarer behaviours, there was no mention of how this could affect the data in the discussion, consider including a mention of how this could affect the data for a more objective point of view.

-The chosen minimal body-part scheme (trunk/legs/head) while I understand that it's to make it simpler, it creates room for potential loss of nuance, some behaviours might be better captured by including additional body parts or graded/intensity measures.

- Different clustering methods gave partly inconsistent groupings for some categories, while the triangulation approach is a strength, it also highlights that clustering solutions depend on method choice and parameter settings which could mean this is not repeatable.

- Although the authors state collars were tested and found not to alter behaviour, any tagging may alter detection or subtle behaviours that are more subtle and this risk was not discussed.

- the ethogram and video examples could be used for standardised coder training and inter-observer reliability testing before new studies - this might have been done and if so I would like to see a brief mention of this as it would add reliability to the tests.

Overall with some key alterations I believe this has a lot of potential.

**Do you want your identity to be public for this peer review?** For information about this choice, including consent withdrawal, please see our Privacy Policy

Reviewer #1: No

You may also use PLOS’s free figure tool, NAAS, to help you prepare publication quality figures: https://journals.plos.org/plosone/s/figures#loc-tools-for-figure-preparation

---

## [Author Response · Author response to Decision Letter 1]

3 Feb 2026

Reviewer #1: - Throughout the manuscript there are multiple uses of first person narrative, consider changing this to third person.

We thank the reviewer for the suggestion. While we considered revising the manuscript to third person, we elected to retain the first-person narrative, as it reflects modern academic writing practices and improves clarity and readability.

- The use of excessive wording at times obscures the main point and detracts from the scientific professionalism of the text. Simplifying the language would improve both clarity and impact and help maintain a professional and academic style.

We thank the reviewer for this constructive comment. In response, we have revised the manuscript to simplify the language in several sections, with the aim of improving clarity, conciseness, and overall readability while maintaining a professional academic tone.

- Several constellations (social vs affiliative vs foraging vs vocalisations) overlap across contexts and did not cluster cleanly, indicating limits to context-only categorical assignment for multifunctional behaviours.

This is true. We explained in the Discussion that “… the social and affiliative categories exhibited considerable overlap, making classification more complex, and the clustering methods provided different results for them. By using the results from the three approaches, we were able to assign each constellation to a single behavioural category with greater confidence.” Also, “…it is difficult to assign some of the constellation to a single category because they appear at different contexts (e.g., approach behaviour can be seen in both mating and social behavioural categories/contexts).”

We mentioned in the Discussion several behaviours, such as vigilance, with a potentially subjective interpretation (e.g., keeping a careful watch) that we omitted. However, our ethogram includes the static behaviour stand with the head in a parallel to the ground position, which describes the same mechanical behaviour without the context-related interpretation. Human perception is often biased by the tendency to assign meaning to behaviours that we deem meaningful. This can lead to over-interpretation and assignment of significance to actions in an unobjective manner.

- The presence/absence binary coding per session downplays intensity, duration, and frequency differences between behaviours within sessions, potentially meaning important variations get lost.

Our aim was to cluster the various behaviors by calculating the frequency each pair of categories appears together at each observation event (i.e., distance matrix). Although some other measures like duration and time between behaviors were extracted from the videos, these were irrelevant for the occurrence analysis and are not included in the current study.

-Videos were relatively short on average and sessions ended when focal individuals moved out of view, which can bias detection of longer-duration states or rarer behaviours, there was no mention of how this could affect the data in the discussion, consider including a mention of how this could affect the data for a more objective point of view.

This is part of the reason why we did not use duration for the analysis (i.e., in the above comment). Since we are observing wild animals in a complex setting, at times they move out of sight and the duration can be biased. Thus, in our mind, an occurrence analysis of behaviours observed is more objective.

-The chosen minimal body-part scheme (trunk/legs/head) while I understand that it's to make it simpler, it creates room for potential loss of nuance, some behaviours might be better captured by including additional body parts or graded/intensity measures.

The categories that we selected are ones that are easily defined by an observer in the field. Currently, our observation distance and the use of a telescope do not permit us to video all the behaviors observed. Thus, a more detailed analysis involving body-part scheme is maybe an approach that we will explore in the future. We mention in the Discussion that it is possible to define behaviours using different units with higher complexity (e.g., mouth, eyes, nose, tail, and neck movements; Liu et al., 2009; Schleidt et al., 1984).

- Different clustering methods gave partly inconsistent groupings for some categories, while the triangulation approach is a strength, it also highlights that clustering solutions depend on method choice and parameter settings which could mean this is not repeatable.

That is true. Behaviour is hard to group, especially affiliative categories. That is why we applied all three methods.

- Although the authors state collars were tested and found not to alter behaviour, any tagging may alter detection or subtle behaviours that are more subtle and this risk was not discussed.

Research introduces disturbances, we agree. However, after 26 years of study of the collared population, the aim of this study was to define the behaviours that we can detect under the setup of our long-term tagging.

- the ethogram and video examples could be used for standardised coder training and inter-observer reliability testing before new studies - this might have been done and if so I would like to see a brief mention of this as it would add reliability to the tests.

Exactly! We will be using the videos and ethogram to train new researchers. Thank you for suggesting that we add it to the manuscript.

Overall with some key alterations I believe this has a lot of potential.

Thank you!

---

## [Editor Report · Decision Letter 1]

4 Feb 2026

Developing objective tools to study rock hyrax (Procavia capensis) behaviour in the field

PONE-D-25-41330R1

Dear Dr. Koren,

We’re pleased to inform you that your manuscript has been judged scientifically suitable for publication and will be formally accepted for publication once it meets all outstanding technical requirements.

Kind regards,

James Edward Brereton, MSc

Academic Editor

PLOS One
---

## [Editor Report · Acceptance letter]

PONE-D-25-41330R1

PLOS One

Dear Dr. Koren,

I'm pleased to inform you that your manuscript has been deemed suitable for publication in PLOS One. Congratulations! Your manuscript is now being handed over to our production team.

Kind regards,

on behalf of

Mr. James Edward Brereton

Academic Editor

PLOS One